# miR-140-5p Attenuates Hypoxia-Induced Breast Cancer Progression by Targeting Nrf2/HO-1 Axis in a Keap1-Independent Mechanism

**DOI:** 10.3390/cells11010012

**Published:** 2021-12-22

**Authors:** Megharani Mahajan, Sandhya Sitasawad

**Affiliations:** Redox Biology Laboratory, National Centre for Cell Science (NCCS), Pune 411007, India; mmegha@nccs.res.in or

**Keywords:** breast cancer, miR-140-5p, Nrf2, angiogenesis, metastasis

## Abstract

Hypoxia and oxidative stress significantly contribute to breast cancer (BC) progression. Although hypoxia-inducible factor 1α (Hif-1α) is considered a key effector of the cellular response to hypoxia, nuclear factor erythroid 2–related factor 2 (Nrf2), a master antioxidant transcription factor, is a crucial factor essential for Hif-1α-mediated hypoxic responses. Hence, targeting Nrf2 could provide new treatment strategies for cancer therapy. miRNAs are potential regulators of hypoxia-responsive genes. In a quest to identify novel hypoxia-regulated miRNAs involved in the regulation of Nrf2, we found that miR-140-5p significantly affects the expression of Nrf2 under hypoxia. In our study, miR-140-5p expression is downregulated in BC cells under hypoxic conditions. We have identified Nrf2 as a direct target of miR-140-5p, as confirmed by the luciferase assay. Knockdown of miR-140-5p under normoxic conditions significantly enhanced Nrf2/HO-1 signaling and tumor growth, angiogenesis, migration, and invasion in BC. In contrast, overexpression of miR-140-5p under hypoxic conditions revealed opposite results. Further silencing Nrf2 expression mimicked the miR-140-5p-induced anti-tumor effects. Consistent with the knockdown of miR-140-5p in vitro, mice injected with miR-140-5p-KD cells exhibited dramatically reduced miR-140-5p levels, increased Nrf2 levels, and increased tumor growth. In contrast, tumor growth is potently suppressed in mice injected with miR-140-5p-OE cells. Collectively, the above results demonstrate the importance of the Nrf2/HO-1 axis in cancer progression and, thus, targeting Nrf2 by miR-140-5p could be a better strategy for the treatment of Nrf2-driven breast cancer progression.

## 1. Introduction

The tumor microenvironment plays a crucial role in therapeutic resistance and cancer progression. Hypoxia is an integral component of tumor microenvironment that promotes angiogenesis and metastasis thus, representing a significant barrier to effective cancer treatment strategy [1,2]. Hif-1α is the critical player in cell response to hypoxia which triggers transcriptional program that allows them to survive under the harsh tumor microenvironment. Hif-1α is a prolyl hydroxylase domain-containing protein (PHDs), post-translationally regulated by reactive oxygen species (ROS). Hypoxia leads to ROS generation, causing oxidative stress, which is known to activate the transcription of genes involved in promoting angiogenesis and altering cellular metabolism [3,4]. In response to oxidative stress, Nrf2 is activated, which regulates Hif-1α. Recent studies have shown that hypoxia translocates Nrf2 into the nucleus and increases the expression of both HO-1 and Hif-1α via PI3K/Akt signaling pathway [5]. Crosstalk between Hif-1α and Nrf2 is essential for tumor cell survival and progression [4]. Besides ROS, Hif-1α and Nrf2 signaling are linked by cellular context and interact to promote metastasis and chemo or radioresistance. Growing evidence also supports the role of Nrf2 in activating and maintaining the Hif-1α response as silencing Nrf2 reduces Hif-1α level [6,7]. In addition, a recent study showed elevated levels of Nrf2 in BC tissue and breast cancer stem cells, further confirming its cancer-promoting roles [8,9,10]. Thus, targeting Nrf2 could be a promising chemotherapeutic strategy for treating solid tumors.

MicroRNAs (miRNAs) are now known to regulate several physiological and pathological conditions, including hypoxia and oxidative stress. Since miRNAs can orchestrate cellular redox homeostasis, these are considered central players in regulating cancer [11,12]. Earlier reports suggest that one of the miRNAs, miR-140-5p, is known to regulate Nrf2. However, its role in regulating breast tumor angiogenesis and metastasis in response to hypoxia needs to be evaluated. Several cancers, including breast [13,14,15], gastric [16,17], lung [18], multiple myeloma [19], retinoblastoma [20], laryngeal squamous cell carcinoma (LSCC) [21], and nephroblastoma [22], have shown reduced expression of miR-140-5p which played a crucial role in cancer progression. In the current study, we observed significant repression of miR-140-5p in BC cells under hypoxia. For the first time, we have shown hypoxia-mediated regulation of miR-140-5p and its involvement in regulating tumor angiogenesis and metastasis in BC. Detailed investigation of the mechanism revealed that Nrf2 is the authentic target of miR-140-5p. Our studies showed that miR-140-5p significantly inhibited hypoxia-mediated angiogenesis and metastasis by controlling Nrf2/HO-1 axis. Our finding highlights the importance of suppressing miR-140-5p under hypoxia conditions by promoting tumor progression and thus conferring an aggressive phenotype to hypoxic tumor cells.

## 2. Materials and Methods

### 2.1. Bioinformatics Analysis

To search for potential miRNAs regulating 3′-UTR (untranslated region) of Nrf2 mRNA, we used three different databases such as Target Scan [23], miRDB [24], and miRSystem [25]. The predicted miRNAs by all three databases are shown in Appendix A. The expression of overlapping miRNAs from all three databases was confirmed by qRT-PCR (quantitative reverse transcription polymerase chain reaction).

### 2.2. Cell Culture and Hypoxia Treatment

BC cell lines (MCF-7 and MDA-MB-231) were obtained from ATCC and maintained at National Centre for Cell Science (NCCS), Pune, India. Cell lines were maintained in DMEM plus 10% FBS (Gibco, Rockville, MD, USA) and PenStrep (100 units/mL penicillin and 100 µg/mL streptomycin). For hypoxia experiments, 70% of confluent cells were cultured in plain DMEM under hypoxic conditions (1% O_2_) for 48 h (Billups-Rothenberg, Del Mar, CA, USA).

### 2.3. Vector Construction

The sequence of oligonucleotides used to generate miRNA overexpression or knockdown cassettes and Nrf2 3′-UTR constructs was listed in Appendix A. Small hairpin-based miR-140-5p overexpression (miR-140-5p-OE) construct was designed as described previously [26], and tough decoy-based knockdown construct (miR-140-5p-KD) was developed as described previously [27] by cloning in pLKO.1 – TRC vector (addgene no. #10878) into the AgeI/EcoRI site. The empty pLKO.1 – TRC (EV) vector was used as control.

The shNrf2 lentivirus plasmid was a kind gift from Dr. Mahadeb Pal. Plasmid encoding scrambled shRNA (SC) as a negative control was purchased from addgene (addgene no. #1864).

For the 3′-UTR luciferase construct, wildtype (WT) or mutant (MUT) miR-140-5p response element in 3′-UTR of Nrf2 mRNA was cloned into pMIR-REPORT luciferase vector (Ambion, Austin, TX, USA) between XhoI and NotI site. Positive constructs were verified by DNA sequencing.

### 2.4. Lentivirus Production, Transduction, and Generation of Stable Cell Lines

Lentiviral particles were produced by co-transfecting transfer plasmid (miR-140-5p-OE, miR-140-5p-KD, or shNrf2 (Nrf2-KD)) along with the psPAX2 (Addgene plasmid # 12260) and the pMD2.G (Addgene plasmid # 12259). 1.8 million 293FT (Thermo Fisher Scientific, Carlsbad, CA, USA # R70007) cells were seeded in 100 mm dishes. The cells were transfected using Lipofectamine 2000 (Thermo Fisher Scientific # 11668019). Total of 15 µL of Lipofectamine 2000, 4 µg of transfer plasmid, 2.8 µg of psPAX2, and 1.2 µg of pMD2.G were used per transfection. After 6 h, the transfection medium was replaced with harvest medium (DMEM supplemented with 30% FBS, 1X NEAA, and antibiotics). Lentivirus-containing medium was harvested after 48 h and used for transduction.

EV, miR-140-5p-OE, miR-140-5p-KD, SC, or Nrf2-KD stable cells were established by transducing MDA-MB-231 cells with respective lentiviruses followed by selection in 2 ug/mL puromycin for two weeks. The overexpression and knockdown efficiency was confirmed by qRT-PCR.

### 2.5. Preparation of Conditioned Medium (CM)

Conditioned media (CM) were collected from MDA-MB-231 miR-140-5p-OE, miR-140-5p-KD, EV, SC, or shNrf2 cells by growing in basal DMEM under normoxia or hypoxia for 48 h.

### 2.6. RNA Isolation, cDNA Synthesis, and qRT-PCR

Total RNA was extracted by the Trizol method (Invitrogen, Carlsbad, CA, USA). About 1 ug of total RNA was used for cDNA synthesis using verso cDNA synthesis kit (Thermo Scientific, Walthman, MA, USA). All primers were obtained from IDT, and the sequences of same are provided in Appendix A.

The Stem-Loop (SL) RT primers and real-time PCR primers for miRNA were designed as described previously [28]. The miRNAs were reverse transcribed into cDNAs using SuperScript III reverse transcription kit (Invitrogen, Carlsbad, CA, USA) [29].

qRT-PCR was performed by using QuantStudio 3 (Applied Biosystems, Foster City, CA, USA). mRNA and miRNA levels were detected using Power Up™ SYBR™ Green Master Mix (Applied Biosystems, Austin, TX, USA). GAPDH and U6 SnRNA were used as the internal control for detecting mRNA and miRNA, respectively.

### 2.7. Western Blotting

Western blotting was performed as described previously [30], using the antibodies described in Appendix A. The Substrate Detection Kit detected protein–antibody complexes (Thermo Fisher, Carlsbad, CA, USA). GAPDH or β-tubulin was used as a loading control.

### 2.8. Luciferase Assay

MDA-MB-231 cells with stable miR-140-5p-OE or miR-140-5p-KD were seeded in 24 well plates (5 × 10^4^ cells/well). Cloned pMIR-REPORT vector (400 ng) was co-transfected along with EGFP endogenous control (50 ng) using Lipofectamine 2000 reagent (Invitrogen) as per manufacturer’s instructions. After 48 hypoxia, cells were lysed in passive lysis buffer (Promega, Madison, WI, USA). The luciferase activities were measured using the Dual-Luciferase^®^ Reporter Assay System (Promega, Madison, WI, USA). The firefly luciferase activity was normalized by EGFP (renilla luciferase) control vector.

### 2.9. MTT, Colony Formation and Apoptosis Assay

MTT and colony formation assays were performed as described previously [31]. For colony formation assay, 200 cells were seeded per well in a 6-well plate and incubated at 37 °C for 20 days. The colonies were fixed (3.7% PFA), stained with 0.2% crystal violet, photographed, and counted. Colony-forming unit (CFU) was calculated as follows: CFU = (no. of colonies/no. of cells seeded) × 100 [32]. Apoptosis assay was performed as described previously [10].

### 2.10. Angiogenesis Assays

Rat aortic ring assay [33] and yolk sac membrane (YSM) assay [34] were performed as described previously to evaluate the angiogenic potential of miR-140-5p using CM from EV, miR-140-5p-KD, or miR-140-5p-OE cells. The experiments were repeated three times. Images of the YSM assay were taken using a stereomicroscope (Nikon, Tokyo, Japan).

The secretion of pro-angiogenic factors was evaluated by ELISA Kit (R&D Systems, Minneapolis, MN, USA) as per the manufacturer’s instructions.

### 2.11. Migration/Invasion/Zymography Assays

Horizontal movement ability by performing scratch wound healing assay and vertical movement ability by performing transwell migration and invasion assay was performed as described previously [35]. Gelatin zymography was performed as described previously [36] to analyze MMP-9 and MMP-2 activity using CM derived from miR-140-5p-KD or miR-140-5p-OE cells.

### 2.12. In Vivo Study

For all animal studies, ethical approval was obtained from Institutional Animal Ethics Committee (NCCS). NOD/SCID (Female) mice of age 6–8 weeks old were used for the tumorigenicity assay. MDA-MB-231 cells (2 × 10^6^) with stable miR-140-5p-KD, miR-140-5p-OE, or EV were administered subcutaneously into the mammary fat pad of NOD/SCID mice along with matrigel (BD BioSciences, Bedford, MA, USA) in a 1:1 ratio (*n* = 5). Mice were observed regularly for papable tumors. Tumor growth was recorded every third day using the Vernier caliper. According to the institutional ethical guidelines, all the mice were euthanized, and tumors were collected for further study. Tumor volume was calculated as follows: (Length × Width2)/2.

### 2.13. Immunohistochemistry (IHC)

IHC staining was performed as described previously [37] using antibodies against E-cadherin (BD Biosciences, San Jose, CA, USA), β-catenin (Santa cruz, Dallas, TX, USA), Twist (Sigma-Aldrich, St. Louis, MO, USA), and Slug (Abcam, Burlingame, CA, USA). The immunoreaction was observed under a bright-field microscope at a 60× objective lens.

### 2.14. Statistical Analysis

Data were expressed as the mean ± standard deviation (SD). The results were from at least three independent experiments. Statistical comparisons were made between two groups with the *t*-test and between multiple groups by ANOVA. Prism software (GraphPad, San Diego, CA, USA), was used to analyze statistical significance. A value of *p* < 0.05 was considered statistically significant.

## 3. Results

### 3.1. Hypoxia Suppresses miR-140-5p Expression, and It Inversely Correlates with Nrf2 Levels

Through in silico analyses as described in materials and methods, we primarily identified potential candidate miRNAs involved in regulating Nrf2 using three different algorithms (Appendix A) and selected four miRNAs with target sites present in all three algorithms as candidate miRNAs (Figure 1A,B). In order to identify the miRNAs regulated by hypoxia, we subjected BC cells to hypoxic (1% oxygen) or normoxic (20% oxygen) conditions. Hypoxia significantly increased the expression of Hif-1α in breast cancer cells (Appendix A). The expression of the four miRNAs was further confirmed by qRT-PCR and found that miR-140-5p expression level was reduced in both the BC cell lines under hypoxic conditions (Figure 1C). We also analyzed the expression of miR-140-5p in publicly available breast cancer TCGA datasets using the ULCAN. miR-140-5p expression is significantly attenuated in primary breast tumors as compared to normal breast tissues. In addition, patients with low miR-140-5p expression had shorter overall survival when compared to patients with high miR-140-5p expression (Figure 1D). To further understand the association between miR-140-5p and Nrf2 under hypoxic conditions, we first explored the effects of hypoxia on Nrf2 expression. The results demonstrated that Nrf2 expression in BC cells was significantly increased when exposed to hypoxia. However, hypoxia treatment non-significantly affected the keap1 levels, a negative regulator of Nrf2, while HO-1 level was significantly increased in both the BC cells under hypoxia (Figure 1E,F). Collectively, the above data indicate repression of miR-140-5p and its inverse correlation with Nrf2 expression in BC cells under hypoxic conditions.

### 3.2. MiR-140-5p Directly Targets Nrf2 in BC

To confirm the in silico prediction, we initially designed miR-140-5p loss-of-function and gain-of-function experiments. The efficiency of miR-140-5p knockdown or overexpression is confirmed by examining its relative expression by qRT-PCR (Appendix A). Of note, qRT-PCR and Western blotting results showed that knockdown of miR-140-5p under normoxia elevated Nrf2 at both mRNA and protein levels, while overexpression of miR-140-5p under hypoxia reversed this effect. We did not detect a significant change in the keap1 mRNA and protein levels in both miR-140-5p-KD and miR-140-5p-OE cell lines. However, miR-140-5p knockdown under normoxia increased HO-1 mRNA and protein levels, while miR-140-5p overexpression under hypoxia revealed an opposite effect (Figure 2A–D). We further inhibited Nrf2 expression to study its effect on miR-140-5p expression level. The results indicated that inhibiting Nrf2 increased miR-140-5p expression in BC cells under hypoxic conditions (Figure 2E). To further confirm miR-140-5p binds directly to Nrf2 mRNA, we performed luciferase assay by constructing WT or MUT Nrf2 3′-UTR reporter vector (Figure 2F) and transfected into miR-140-5p-OE and miR-140-5p-KD MDA-MB-231 cells. Overexpression of miR-140-5p significantly reduced the luciferase activity of the WT Nrf2 3′-UTR but did not alter the luciferase activity of the MUT Nrf2 3′-UTR. As a control experiment, we have also done the same experiment in the miR-140-5p knockdown condition. However, miR-140-5p knockdown did not cause any change in luciferase activity associated with WT or MUT Nrf2 3′-UTR reporter vector (Figure 2G). Collectively, these findings indicate that miR-140-5p directly targets Nrf2 3′-UTR and negatively regulates its gene expression.

### 3.3. Interplay between miR-140-5p and Hif-1α

We next sought to understand the mechanism underlying miR-140-5p repression under hypoxia. Since Hif-1α is the master hypoxia-responsive transcription factor, we thus sought to study their involvement in miR-140-5p regulation. We transiently inhibited the level of Hif-1α using shRNA vector and scored for its effect on miR-140-5p levels. Expression of Hif-1α decreased dramatically at mRNA level as confirmed by qRT-PCR (Figure 3A). However, the expression of miR-140-5p was non-significantly changed after Hif-1α inhibition (Figure 3B), suggesting that miR-140-5p is regulated independently of Hif-1α under hypoxia.

Given that miR-140-5p is regulated independently of Hif-1α under hypoxia, we next assessed the involvement of miR-140-5p in the regulation of Hif-1α expression. MDA-MB-231 cells with stable miR-140-5p knockdown under normoxia or overexpression under hypoxia were subjected to flow cytometric +analysis, and Hif-1α expression was analyzed. Interestingly, knockdown of miR-140-5p under normoxia significantly induced the Hif-1α expression (Figure 3C). On the other hand, overexpression of miR-140-5p under hypoxia reduced the Hif-1α level (Figure 3D). However, no specific miR-140-5p-binding sites in the 3′-UTR of Hif-1α mRNA were predicted by the online databases, suggesting that the regulation of Hif-1α activity by miR-140-5p might be indirect. We, therefore, searched for a transcription factor that has the potential to regulate Hif-1α expression and, at the same time, could be a target for miR-140-5p. The result revealed that Nrf2 had a confirmed binding site for miR-140-5p in their 3′-UTR and held the potential to regulate Hif-1α expression.

### 3.4. miR-140-5p Inhibits Cell Viability, Proliferation, and Colony Formation under Hypoxia

Hypoxia is associated with tumor aggressiveness because of its role in promoting cell growth, migration, and invasion [38]. Therefore, we next examined whether hypoxia-regulated miR-140-5p is involved in MDA-MB-231 cell growth, angiogenesis, migration, and invasion. Since MCF-7 cells usually do not migrate or invade [39], we performed further functional experiments using only MDA-MB-231 cells. Initially, we tested the functional significance of hypoxia-regulated miR-140-5p in cell growth, proliferation, and colony formation. Knocking down miR-140-5p under normoxia promoted cell viability, proliferation ability, and colony formation, while overexpression of miR-140-5p under hypoxia revealed an opposite effect (Figure 4A–E). We then looked for the involvement of miR-140-5p in cell adhesion to different extracellular matrices (ECMs). Although MDA-MB-231 cells with miR-140-5p knockdown under normoxia did not affect cell adhesion to different ECM components, miR-140-5p overexpression under hypoxia significantly reduced cell adhesion on fibronectin, laminin, and collagen IV (Appendix A). Collectively, above results suggest that miR-140-5p inhibits cell viability, proliferation, and colony formation of BC cells under hypoxic condition.

### 3.5. miR-140-5p Inhibits Angiogenesis in BC

As hypoxia is an important factor to induce angiogenesis, we predicted that hypoxia-regulated miR-140-5p could also regulate angiogenesis. To define the correlation between miR-140-5p/Nrf2 and angiogenesis, we initially examined the secretion of different pro-angiogenic factors (IL-8, VEGF, and bFGF) by ELISA. CM from miR-140-5p-KD cells under normoxia showed a significant increase in the secretion of IL-8 and VEGF, while there was no change in the level of bFGF. The CM from miR-140-5p-OE cells under hypoxia revealed an opposite effect with no change in the level of bFGF. Further experiments showed that knockdown of Nrf2 under hypoxia mimicked the effects of miR-140-5p overexpression regarding the secretion of different pro-angiogenic factors (Figure 5A).

Based on the above observation, we next evaluated the angiogenic ability of miR-140-5p using rat aortic ring assay. Our data show that the CM from miR-140-5p-KD cells under normoxia markedly induced angiogenesis in the rat aortic rings compared to EV control (Figure 5B). In contrast, CM from miR-140-5p-OE cells under hypoxia revealed an opposite effect (Figure 5C).

As the mature miR-140-5p sequence is conserved across chicken, mouse, and human species (Appendix A), this supports the translation of our in vitro data to in vivo test for angiogenic testing miR-140-5p. Therefore, we performed In Ovo YSM assay to confirm the angiogenic potential of miR-140-5p. Our data show that the CM from miR-140-5p-KD cells under normoxia markedly induced angiogenesis in the chicken YSMs compared to EV control. Moreover, CM from miR-140-5p-OE cells revealed opposite results (Figure 5D,E). To elucidate the downstream effectors involved in miR-140-5p-regulated angiogenesis, the expression level of angiogenic factors, VEGF and COX-2 were investigated. VEGF expression was increased in miR-140-5p-KD cells under normoxia or decreased in miR-140-5p-OE cells under hypoxia. In addition, knockdown of Nrf2 under hypoxia reduced the VEGF expression consistent with the results obtained from miR-140-5p overexpression under hypoxic conditions (Figure 5F). However, the levels of COX-2 were non-significantly altered in MDA-MB-231 cells with miR-140-5p-KD or miR-140-5p-OE (Appendix A), suggesting that miR-140-5p-regulated angiogenesis did not appear to be through COX-2 signaling. Taken together, we revealed a critical miR-140-5p/Nrf2/HO-1/VEGF signaling in miR-140-5p-mediated angiogenesis.

### 3.6. miR-140-5p Inhibits BC Cell Migration and Invasion

Migration and invasion are the characteristic features of metastatic tumors. Therefore, we evaluated the effect of the miR-140-5p expression on these cellular processes. Initially, the horizontal cellular movement ability of BC cells was assessed by performing a scratch wound-healing assay. As shown in Figure 6, knocking down miR-140-5p under normoxia significantly promoted, whereas its overexpression under hypoxia significantly reduced the motility of BC cells (Figure 6A,B). We then used a transwell assay to investigate the functional role of miR-140-5p on vertical cellular migration and invasion. Compared to the EV, stable miR-140-5p-KD cells under normoxia have significantly higher migration and invasion capacity, while miR-140-5p-OE cells under hypoxia have significantly lower migration and invasion capacity. Moreover, knockdown of Nrf2 under hypoxia significantly reduced the migration and invasion capacity, thus, mimicking the role of miR-140-5p overexpression under hypoxia (Figure 6C–E). To understand the underlying mechanism, we measured the activities of two critical gelatinases, MMP-9 and MMP-2, whose secretion and activities correlate with solid tumors’ invasiveness. The results showed that the activities of both MMP-9 and MMP-2 were significantly increased in miR-140-5p-KD cells under normoxia. Moreover, miR-140-5p-OE cells under hypoxia revealed an opposite effect (Figure 6F). Further, the expression level of vimentin, an EMT marker, was analyzed by Western blot. For miR-140-5p-KD cells under normoxia, vimentin expression was significantly high compared to miR-140-5p-OE cells under hypoxia. In addition, silencing Nrf2 under hypoxia mimicked the effect of miR-140-5p overexpression on vimentin expression (Figure 6G). Collectively, the above results suggest that miR-140-5p controls secretion and activity of MMP’s and vimentin through regulation of Nrf2 expression to inhibit migration and invasion of the BC cell line.

### 3.7. miR-140-5p Promotes BC Cell Apoptosis

To elucidate the mechanism underlying miR-140-5p-induced disruption of cell growth, cell cycle and apoptosis analyses were performed by flow cytometry. Our data revealed that both miR-140-5p-KD cells under normoxia and miR-140-5p-OE cells under hypoxia had a non-significant effect on the cell cycle (Appendix A). Following a judgment of the cell cycle, the proportion of apoptotic cells was examined by AnnexinV/PI double staining. Compared with EV, the proportion of apoptotic cells was significantly reduced in miR-140-5p-KD cells under normoxia. In addition, miR-140-5p-OE cells under hypoxia revealed opposite results (Figure 7A). To delineate the underlying mechanism, we examined the expression levels of key apoptotic proteins by Western blot. The results demonstrated that miR-140-5p knockdown under normoxia significantly decreased the levels of pro-apoptotic proteins (PARP, Bid, and BAX) and increased the level of anti-apoptotic protein (Bcl2) (Figure 7B). On the other hand, overexpression of miR-140-5p under hypoxia reversed this effect (Figure 7C). Furthermore, silencing Nrf2 promoted the expression of pro-apoptotic proteins and reduced the level of anti-apoptotic proteins (Figure 7D). Since the ratio of Bcl2/BAX has a key role in apoptosis (low Bcl2/BAX ratio indicates more apoptosis and high Bcl2/BAX ratio indicates less apoptosis), we evaluated changes in the Bcl2/BAX ratio. Silencing miR-140-5p under normoxia significantly increased the Bcl2/BAX ratio (Figure 7E). However, overexpression of miR-140-5p under hypoxia decreases the Bcl2/BAX ratio (Figure 7F), and silencing Nrf2 under hypoxia also decreases the Bcl2/BAX ratio (Figure 7G), further supporting the above result. Taken together, these results indicated that miR-140-5p activated apoptotic signaling pathway and functioned as a suppressor of breast tumor growth, at least in part, by suppressing Nrf2.

### 3.8. MiR-140-5p Inhibits Breast Tumor Growth in a Mouse Model

To confirm the function of miR-140-5p in vivo, EV, miR-140-5p-KD, or miR-140-5p-OE MDA-MB-231 cells were subcutaneously administered into the mammary fat pad of NOD/SCID mice. Once palpable tumor developed, tumor volumes were measured every 3rd day. After six weeks, mice were sacrificed, xenografts were collected and weighed. Representative images of mice harboring MDA-MB-231 xenografts were presented. Compared with the EV, the size of tumors from the miR-140-5p knockdown group was increased (Figure 8A,B). Moreover, tumor weight and volume were higher in the miR-140-5p-KD group than in the EV group (Figure 8C). Further, tumors from both the group were analyzed for miR-140-5p and Nrf2 expression, and it was found that compared with EV group, the miR-140-5p level was reduced while Nrf2 level was increased in miR-140-5p knockdown tumors which is consistent with our in vitro results (Figure 8D). Next, we performed an IHC analysis of EMT markers in the xenograft tumor tissues. The data demonstrate that E-cadherin and β-catenin expressions were reduced while slug and twist expressions were increased in miR-140-5p-KD xenograft tumors compared to those expressing the EV group (Figure 8E). On the other hand, tumor growth was potently inhibited in the miR-140-5p-OE group when compared to the EV group (Appendix A). These findings indicate that overexpression of miR-140-5p could suppress tumor growth through the regulation of Nrf2 pathway.

## 4. Discussion

Global downregulation of several miRNAs under hypoxia has been reported in cancer [11]. However, most of them have not yet been well characterized. Few reports revealed the involvement of miRNAs in hypoxia-mediated signaling pathways [40]. Nrf2-mediated anti-oxidation is at the heart of the hypoxia-mediated oxidative stress response, which confers resistance to chemo- and radiotherapies leading to cancer progression and metastasis [40]. However, the crosstalk between miRNAs and Nrf2 signaling under tumor hypoxia conditions is unknown. This study utilized computational prediction combined with experimental approaches to identify miR-140-5p as the predominant miRNA regulated by hypoxia. Moreover, miR-140-5p expression was repressed in BC cell lines under hypoxia, which is in agreement with the recent reports in different cancers [13,14,15,16,17,18,19,20,21,22]. Further, we reported that reduced miR-140-5p expression was associated with poor survival of patients with BC. However, the specific role of miR-140-5p in hypoxia-mediated response to oxidative stress still remains unclear. Hence, we aimed to understand the potential role of miR-140-5p in hypoxia-mediated regulation of Nrf2 in BC progression. To our awareness, our study is the first to show the potential role of miR-140-5p in hypoxia-mediated BC progression by regulating Nrf2.

Hypoxia is an integral component of the tumor microenvironment which plays a key role in therapeutic resistance and cancer progression [41]. It drives changes in the gene expression pattern, which confer chemoresistance to the malignant tumors [42]. We have also reported that hypoxia in the tumor microenvironment suppresses miR-140-5p expression in a Hif-1α-independent manner in BC cells. Previously a study by Papathanasiou et al. has reported the role of miR-140-5p promoter methylation in its repression in osteoarthritis by changing the binding affinity of SMAD3 [43]. A similar mechanism of miR-140-5p regulation might exist under tumor hypoxia which needs further investigation.

In chronic diseases like cancer, oxidative stress significantly contributes to the pathogenesis by regulating Nrf2 activity. Increased Nrf2 promotes cell growth and proliferation as well as inhibits cell apoptosis in cancer cells to induce cancer progression. Therefore, blocking Nrf2 activity is considered a potential therapeutic approach for cancer prevention. A potent therapeutic intervention is to target Nrf2 by miRNAs because of their short size, ability to respond rapidly to hypoxic stress, and also to regulate multiple genes and pathways simultaneously. Here, we report a novel mode of Nrf2 regulation by miR-140-5p under hypoxia in breast tumors. We have also shown that knockdown of miR-140-5p under normoxia increases the levels of Nrf2 while overexpression of miR-140-5p under hypoxia reduces the levels of Nrf2 transcript and protein. Further, the luciferase assay confirmed that Nrf2 is a target of miR-140-5p. Our results are in agreement with the previous findings showing increased expression of Nrf2 in BC cells [8,9]. These studies suggest that miR-140-5p might induce Nrf2 mRNA degradation and inhibit its translation. We provide evidence for a new pathway of miRNA-mediated regulation of Nrf2 in a keap1-independent manner. Further functional studies demonstrated that miR-140-5p inhibited BC proliferation and growth.

As part of our research on how miR-140-5p inhibits BC growth, angiogenesis, and metastasis, we showed that Nrf2 and its downstream target HO-1 are major mediators in miR-140-5p-mediated responses, as confirmed by both overexpression and knockdown studies. Previous studies reported a tumor suppressor role of miR-140-5p by regulating VEGF-A expression [13,19]. Here, we report that miR-140-5p regulates VEGF expression through Nrf2/HO-1 axis. We have also shown that VEGF is the major player in miR-140-5p-mediated angiogenic responses. Further, MMP-9, MMP-2, and vimentin are critical components in miR-140-5p-mediated migration and invasion. In addition, Nrf2-knockdown exhibited similar tumor-suppressive roles as observed for miR-140-5p overexpression in BC cells. Collectively, these results confirmed that Nrf2 is an authentic target of miR-140-5p in BC.

Ectopic expression of miR-140-5p potently suppressed the Nrf2-mediated tumor growth in a xenograft mouse model. Moreover, in the xenograft BC model, miR-140-5p expression was reduced that correlates inversely with Nrf2, which is consistent with in vitro results. These results confirm the tumor suppressor property of miR-140-5p under hypoxia.

miR-140-5p acts as a suppressor of tumor growth by inducing apoptosis and hampering proliferation and movement [44,45,46]. In our studies, we have reported that miR-140-5p exhibits tumor-suppressive role by promoting apoptosis and not affecting the cell cycle. Our study significantly broadens the understanding of the regulation of miR-140-5p and supports its role as a tumor suppressor under hypoxia. Even though we observed a significant tumor growth and metastasis in stable miR-140-5p knockdown cells, these effects could be driven by additional factors, and therefore, further investigation is required. Since hypoxia signaling in tumors is a combined effect of multiple cell types, including endothelial cells within the tumor microenvironment, the effect of ectopic expression of miR-140-5p on these cell lines remains to be investigated. Moreover, increased miR-140-5p elevated ROS levels causing oxidative stress by Nrf2/Sirt2/Keap1/HO-1 pathway in mice with atherosclerosis [47]. Since Nrf2 is considered the master regulator of the oxidative stress response, the link between miR-140-5p/oxidative stress/Nrf2 in BC needs to be investigated.

In conclusion, we investigated the effect and mechanism of miR-140-5p in tumor angiogenesis and metastasis under hypoxia. MiR-140-5p is downregulated in BC cells and tissues and functions as a suppressor of angiogenesis and migration and invasion through the silencing of Nrf2 expression. Our data strongly suggest that reduced miR-140-5p plays a vital role in the growth and occurrence of BC. Restoration of miR-140-5p may represent a promising strategy for anti-BC therapy.

## Figures and Tables

**Figure 1 cells-11-00012-f001:**
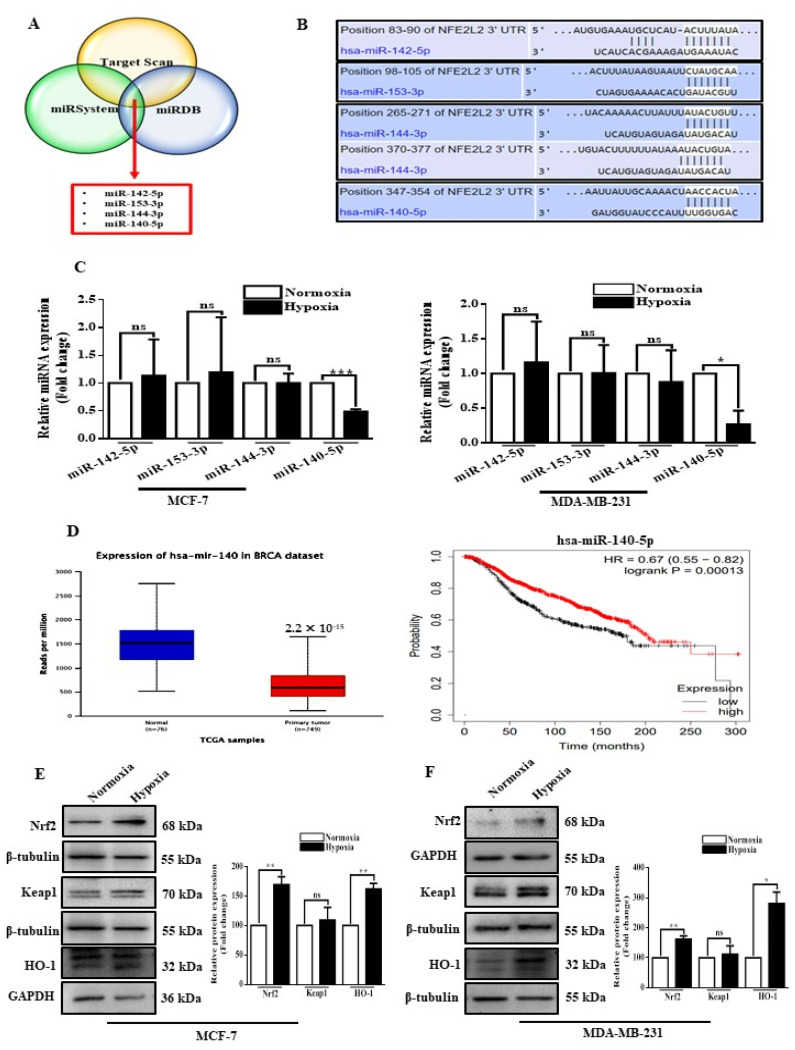
miR-140-5p and Nrf2 expression in BC cells under hypoxia. (**A**) Venn diagram showing common miRNAs computationally predicted to target Nrf2 mRNA by all three algorithms (Target Scan, miRDB, and miRSystem). (**B**) Schematic of the predicted binding site of all four miRNAs in the 3′-UTR of Nrf2 mRNA. (**C**) qRT-PCR analysis of all four selected miRNAs in MCF-7 and MDA-MB-231 under hypoxic conditions. (**D**) Relative miR-140-5p expression in normal (*n =* 76) and primary BC (*n =* 749) samples based on the data from The Cancer Genome Atlas (TCGA) using the ULCAN web. The survival rates of patients with BC having low (Black) or high (red) miR-140-5p expression as estimated by the Kaplan-Meier method using the data from TCGA. (**E**,**F**) Nrf2 and its downstream proteins were analyzed by Western blotting and densitometry analysis using Image J software in MCF-7 and MDA-MB-231. Error bars indicate mean ± SEM (*n =* 3). Student’s *t*-tests were used to compare the means of two groups. ns: not significant, * *p* < 0.05, ** *p* < 0.01 and *** *p* < 0.001 compared to Normoxia).

**Figure 2 cells-11-00012-f002:**
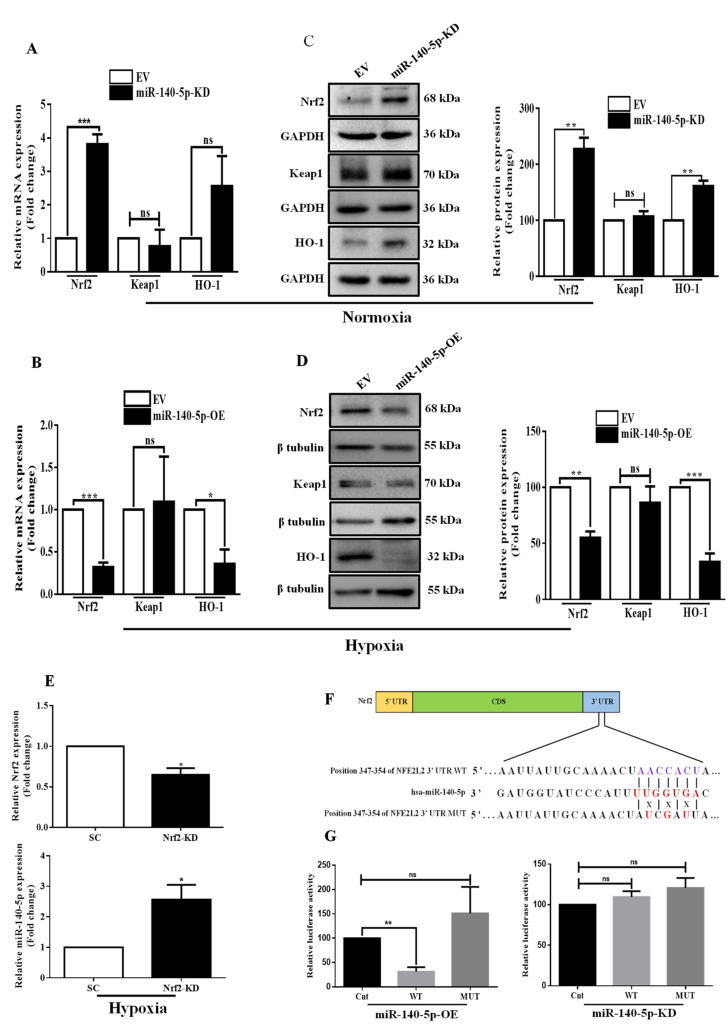
Nrf2 is a direct target of miR-140-5p. (**A**,**B**) The mRNA and (**C**,**D**) protein levels of Nrf2 and its downstream genes were determined by qRT-PCR and Western blotting, respectively, in MDA-MB-231 cells with miR-140-5p-KD under normoxia or miR-140-5p-OE under hypoxia. Quantification of Western blot images was done by image J software. (**E**) MDA-MB-231 cells with stable Nrf2-KD were subjected to hypoxia, and relative Nrf2 mRNA and miR-140-5p levels were measured by qRT-PCR. (**F**) miR-140-5p and its complementary sites in the 3′-UTR of Nrf2 mRNA with WT and MUT sequence are shown. MUT sequence was generated by changing their complementary sites. (**G**) The luciferase assay in MDA-MB-231 cells with either miR-140-5p-OE (left panel) or miR-140-5p-KD (right panel) co-transfected with WT or MUT Nrf2 3′-UTR along with EFGP. EGFP (Renilla luciferase) was used for the normalization of fluorescence intensity. Error bars indicate mean ± SEM (*n =* 3). Student’s *t*-tests were used to compare the means of two groups. ns: not significant, * *p* < 0.05, ** *p* < 0.01, and *** *p* < 0.001 compared to EV.

**Figure 3 cells-11-00012-f003:**
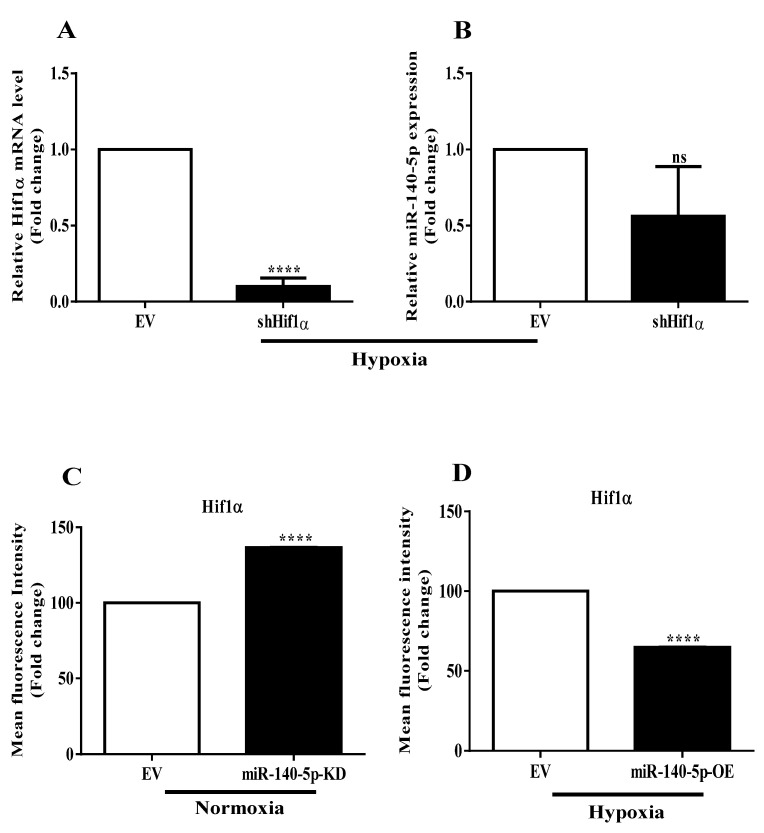
Crosstalk of miR-140-5p and Hif-1α. (**A**) Hif-1α expression was transiently inhibited under hypoxia, and relative Hif-1α mRNA level was measured. (**B**) miR-140-5p expression level was measured in Hif-1α-inhibited cells. (**C**,**D**) Effect of miR-140-5p knockdown under normoxia or overexpression under hypoxia on Hif-1α expression as measured by flow cytometry. Error bars indicate mean ± SEM (*n =* 3). Student’s *t*-tests were used to compare the means of two groups. ns: not significant, **** *p* < 0.0001 compared to EV.

**Figure 4 cells-11-00012-f004:**
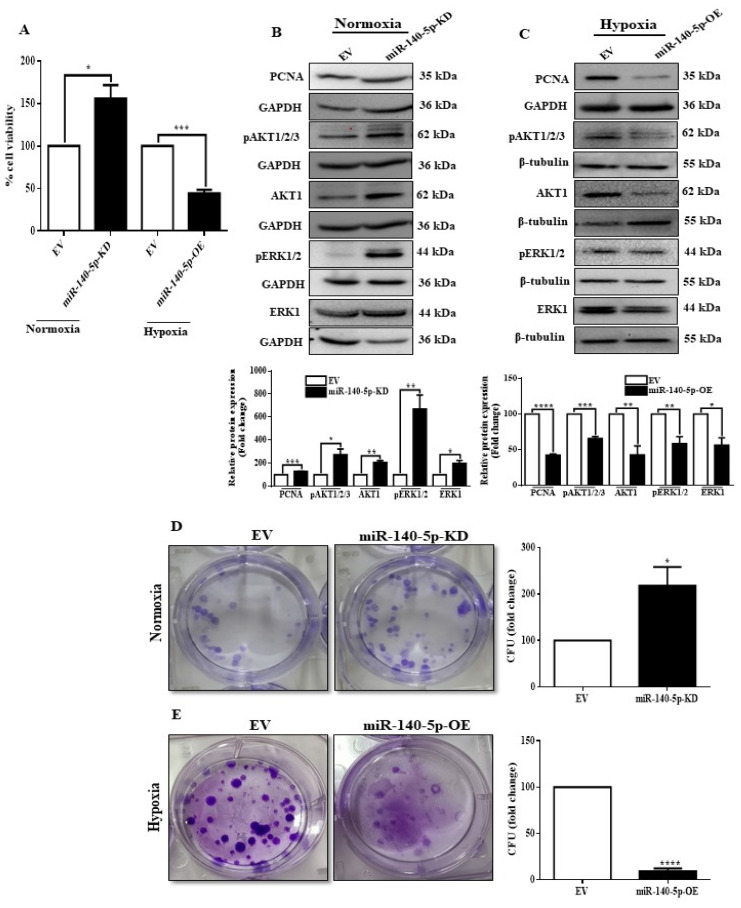
miR-140-5p regulates BC cell growth and colony formation. (**A**) MTT assay to determine cell viability of MDA-MB-231 to examine the effect of miR-140-5p knockdown under normoxia or overexpression under hypoxia. (**B**,**C**) Cell proliferation was assessed by examining proliferation markers by Western blotting after stable miR-140-5p knockdown under normoxia or overexpression under hypoxia. (**D**,**E**) Colony-forming ability was assessed after stable miR-140-5p knockdown under normoxia or overexpression under hypoxia. Error bars indicate mean ± SEM (*n =* 3). Student’s *t*-tests were used to compare the means of two groups. * *p* < 0.05, ** *p* < 0.01, *** *p* < 0.001, and **** *p* < 0.0001 compared to EV.

**Figure 5 cells-11-00012-f005:**
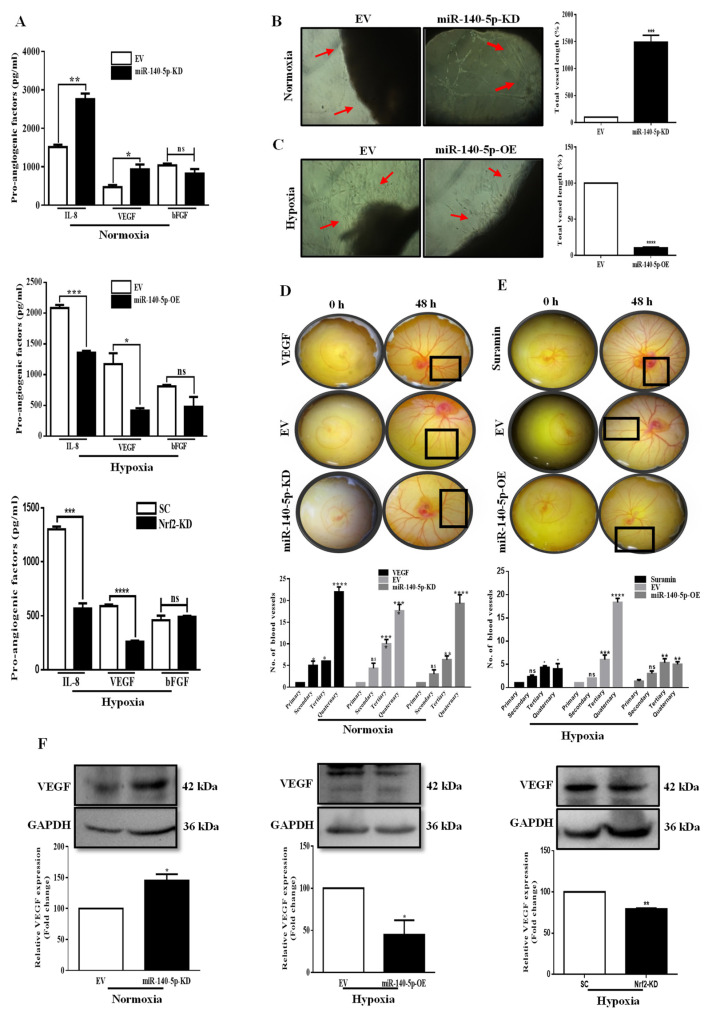
miR-140-5p suppresses angiogenesis in BC. (**A**) ELISA was performed to measure secretion of pro-angiogenic factors (IL-8, VEGF, and bFGF) in CM from miR-140-5p-KD cells under normoxia, miR-140-5p-OE cells or Nrf2-KD cells under hypoxia. (**B**,**C**) Rat aortic ring sections cultured on matrigel containing CM from miR-140-5p-KD cells under normoxia or miR-140-5p-OE cells under hypoxia. The vascularized area is indicated by red arrowheads. Quantification of the vascular area was performed using AngioTool. (**D**,**E**) Representative images of the chick YSM assay treated with CM from either miR-140-5p-KD cells under normoxia or miR-140-5p-OE cells under hypoxia. Here VEGF and Suramin were used as positive and negative controls respectively to test the angiogenic potential of miR-140-5p. (**F**) Analysis of VEGF expression by Western blotting in miR-140-5p-KD cells under normoxia and miR-140-5p-OE cells or Nrf2-KD cells under hypoxia. Error bars indicate mean ± SEM (*n =* 3). Student’s *t*-tests were used to compare the means of two groups. ns: not significant, * *p* < 0.05, ** *p* < 0.01, *** *p* < 0.001, and **** *p* < 0.0001 compared to EV.

**Figure 6 cells-11-00012-f006:**
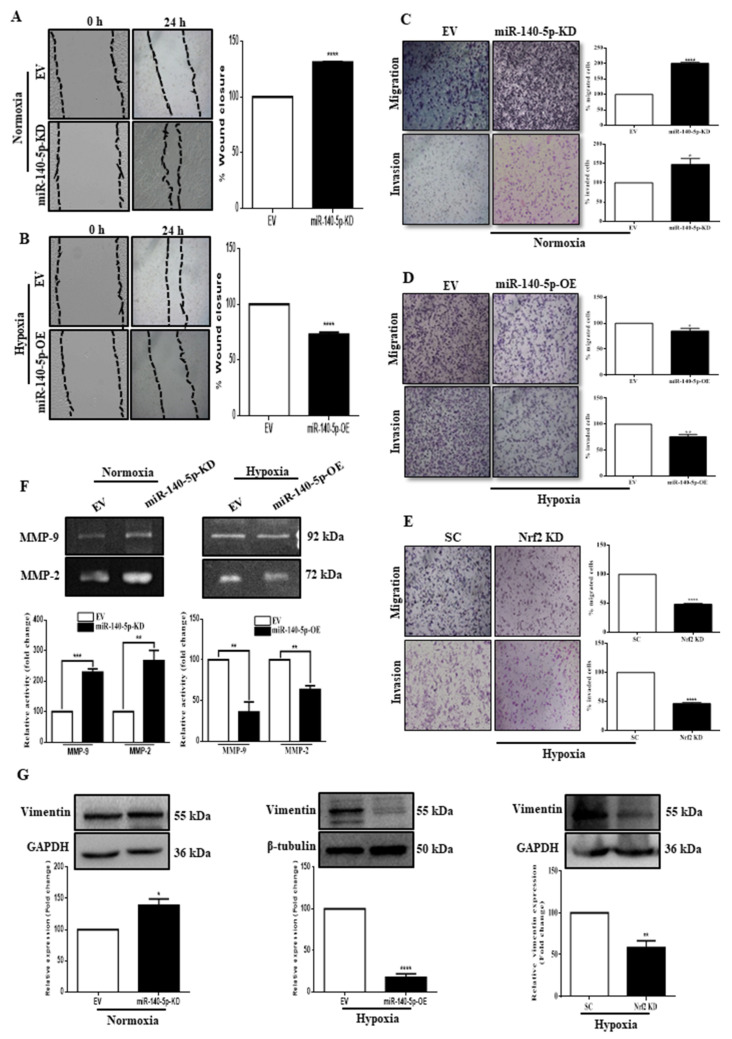
miR-140-5p inhibits BC cell migration and invasion. (**A**,**B**) Scratch wound healing assay was carried out to determine the migration ability of MDA-MB-231 cells with miR-140-5p-KD under normoxia (**A**) or miR-140-5p-OE under hypoxia (**B**) along with their EV control. (**C**–**E**) Transwell migration and invasion assay were performed to determine the migration and invasion ability of MDA-MB-231cells with miR-140-5p-KD under normoxia (**C**), miR-140-5p-OE under hypoxia (**D**), or Nrf2-KD under hypoxia (**E**) along with their controls. (**F**) Gelatin zymography to analyze MMP-9 and MMP-2 activity using CM from miR-140-5p-KD cells under normoxia or miR-140-5p-OE cells from hypoxia. (**G**) Analysis of vimentin expression by Western blotting in miR-140-5p-KD cells under normoxia and miR-140-5p-OE cells or Nrf2-KD cells under hypoxia. Error bars indicate mean ± SEM (*n =* 3). Student’s *t*-tests were used to compare the means of two groups. * *p* < 0.05, ** *p* < 0.01, *** *p* < 0.001, and **** *p* < 0.0001 compared to EV.

**Figure 7 cells-11-00012-f007:**
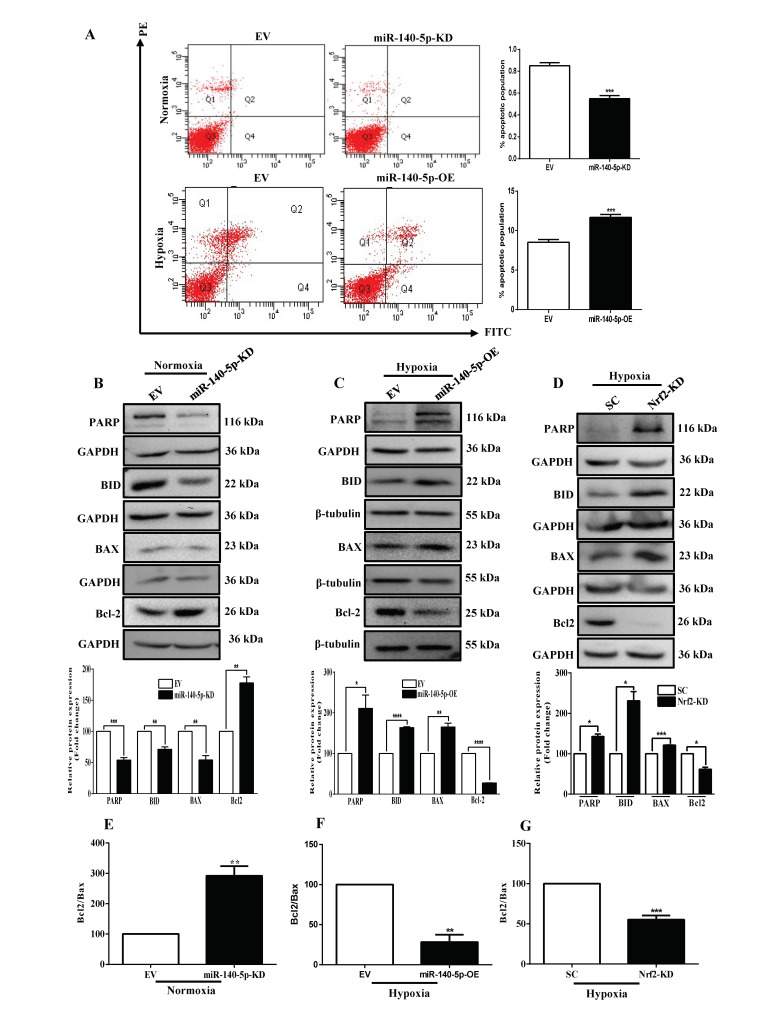
miR-140-5p induces BC cell apoptosis in vitro. (**A**) The effect of miR-140-5p knockdown under normoxia or overexpression under hypoxia on apoptosis as measured by flow cytometry. (**B**–**D**) The levels of apoptotic markers in miR-140-5p-KD cells under normoxia (**B**), miR-140-5p-OE cells under hypoxia (**C**), or Nrf2-KD cells under hypoxia (**D**) as measured by Western blotting and its densitometry by image J software. (**E**–**G**) Bcl-2/Bax ratio after miR-140-5p knockdown under normoxia (**E**), miR-140-5p-OE under hypoxia (**F**), or Nrf2-KD under hypoxia (**G**). Error bars indicate mean ± SEM (*n =* 3). Student’s *t*-tests were used to compare the means of two groups. * *p* < 0.05, ** *p* < 0.01, *** *p* < 0.001, and **** *p* < 0.0001 compared to EV.

**Figure 8 cells-11-00012-f008:**
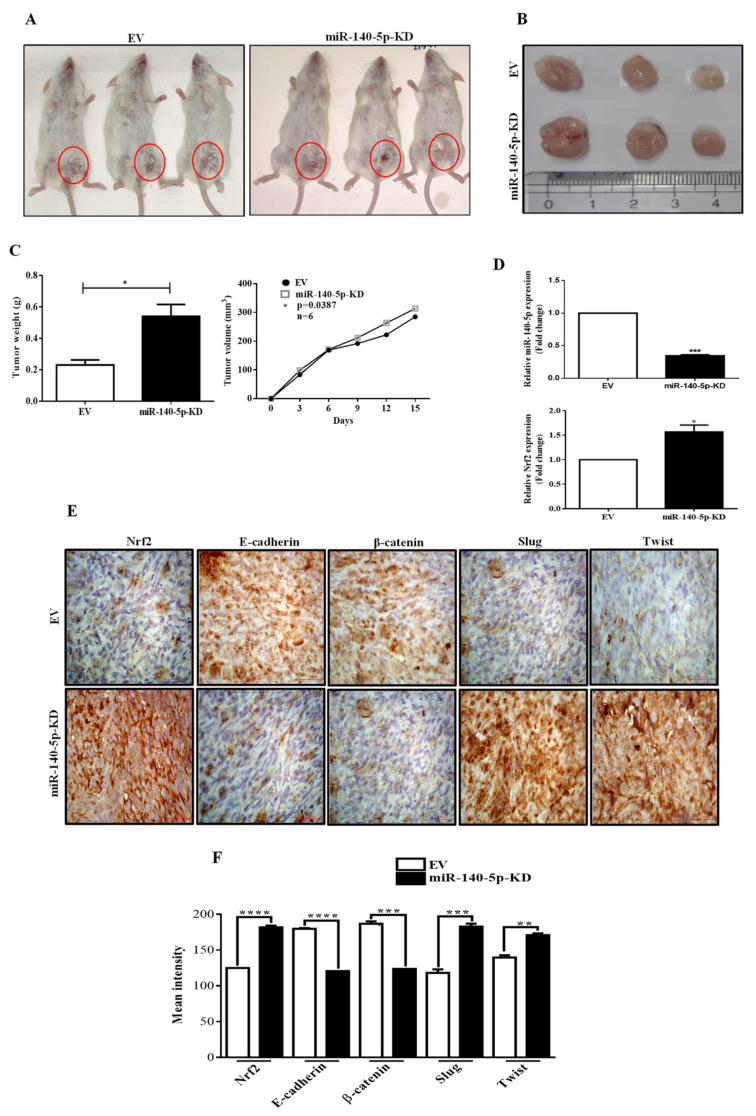
Tumorigenicity assay in a mouse model. (**A**,**F**) EV or miR-140-5p-KD MDA-MB-231 cells were subcutaneously administered into the mammary fat pad of NOD/SCID mice. (**B**) NOD/SCID mice xenograft tumors. (**C**) Average tumor weight and volume (mm^3^) were plotted. (**D**) miR-140-5p and Nrf2 expression in xenograft tumors as measured by qRT-PCR. (**E**) IHC staining of the EMT markers in the xenograft tumors (60x objective lens). Error bars indicate mean ± SEM (*n =* 3). Student’s *t*-tests were used to compare the means of two groups. * *p* < 0.05, ** *p* < 0.01, *** *p* < 0.001, and **** *p* < 0.0001 compared to EV.

## Data Availability

The data presented in this study are available on request from the corresponding author.

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
