# Peer review of "miR-140-5p Attenuates Hypoxia-Induced Breast Cancer Progression by Targeting Nrf2/HO-1 Axis in a Keap1-Independent Mechanism"

_cells, 2021, doi:10.3390/cells11010012_

Round 1

Reviewer 1 Report

In this manuscript by Mahajan et.al., the authors investigate the miR-140-5p regulated Nrf2/HO-1 axis in breast cancer progression. The miR-140-5p was found to be regulated in hypoxia by an unknown mechanism. Further, the authors have identified that mir-140-5p regulates Nrf-2 signaling to control angiogenesis, migration, invasion, and tumor growth in hypoxic conditions. The findings in the manuscript are well presented and identify a relevant pathway. However, there are a few concerns that the authors must address before publication.

  1. The authors demonstrate that the miR-140-5p is downregulated in hypoxia. Can the authors perform the reoxygenation experiment to reinforce the data on hypoxic control of miR-140-5p?
  2. The authors have utilized the miR-140-5p knockdown model in normoxia while the over-expression model is used in hypoxia. While the choice of models is understandable as miR-140-5p is downregulated in hypoxia, the results observed under normoxia conditions may not be relatable to the hypoxic condition. As an alternative, can the authors check the levels of miR-140-5p in multiple BC cell lines and check at least key results in the hypoxic system by making a knock-down in high miR-140-5p expressing BC line.
  3. The miR-140-5p is not found to be regulated by Hif-1a. Hif-2a is another master regulator of hypoxia. Does Hif-2a regulate the expression of miR-140-5p? Additionally, are there any binding sites of miR-140-5p in the Hif2a gene?
  4. The details of statistical analysis are missing from the entire manuscript. The authors should include the relevant statistical test used in each figure legend and include a section on statistical analysis in the material and methods section.
  5. The IHC should be quantified in Fig8E. Also, the authors should perform IHC for Nrf2.
  6. The unprocessed full blots should be provided as supplementary data.
  7. The terms in vitro and in vivo should be italicized in the manuscript.

Author Response

We would like to express our deepest gratitude to the reviewers for their insightful and valuable comments to improve the scientific value of our manuscript. They really emphasized important issues related with our study. Question wise clarification has been given below. 

Reviewer #  1

In this manuscript by Mahajan et.al., the authors investigate the miR-140-5p regulated Nrf2/HO-1 axis in breast cancer progression. The miR-140-5p was found to be regulated in hypoxia by an unknown mechanism. Further, the authors have identified that mir-140-5p regulates Nrf-2 signaling to control angiogenesis, migration, invasion, and tumor growth in hypoxic conditions. The findings in the manuscript are well presented and identify a relevant pathway. However, there are a few concerns that the authors must address before publication.

Thank you very much for your attention to our paper. Based on your suggestions, we have revised the manuscript thoroughly.

Point 1: The authors demonstrate that the miR-140-5p is downregulated in hypoxia. Can the authors perform the reoxygenation experiment to reinforce the data on hypoxic control of miR-140-5p?

Response: We thank the reviewer for the valuable comment. We have examined miR-140-5p expression under hypoxia conditions. However, we have not checked the effect of hypoxia/reoxygenation on miR-140-5p expression, and due to time constrain, it is not possible to perform this experiment now. We will check the expression of miR-140-5p under hypoxia/reoxygenation condition in the future.

Point 2: The authors have utilized the miR-140-5p knockdown model in normoxia while the over-expression model is used in hypoxia. While the choice of models is understandable as miR-140-5p is downregulated in hypoxia, the results observed under normoxia conditions may not be relatable to the hypoxic condition. As an alternative, can the authors check the levels of miR-140-5p in multiple BC cell lines and check at least key results in the hypoxic system by making a knock-down in high miR-140-5p expressing BC line.

Response: We thank the reviewer for the valuable suggestion.  In our study, we have shown that miR-140-5p functions as a tumor suppressor. In most of the experiments, results obtained after knockdown of miR-140-5p under normoxia were reversed in case of overexpression of miR-140-5p under hypoxia except cell adhesion assay in which although knockdown of miR-140-5p under normoxia did not affect cell adhesion to different ECM components, overexpression of miR-140-5p under hypoxia significantly reduced cell adhesion on fibronectin, laminin, and collagen IV. This discrepancy observed in cell adhesion assay might be because of the regulation of some cell adhesion molecules by miR-140-5p.

The reviewer's suggestion on using multiple BC cell lines to check the levels of miR-140-5p is well taken. However, due to time constrain, it is not possible to perform this experiment now. We will certainly perform this experiment in the future.

Point 3: The miR-140-5p is not found to be regulated by Hif-1a. Hif-2a is another master regulator of hypoxia. Does Hif-2a regulate the expression of miR-140-5p? Additionally, are there any binding sites of miR-140-5p in the Hif2a gene?

Response: It is an important question asked by the reviewer. Although Hif-2α is another master regulator of hypoxia, we did not check its involvement in the regulation of miR-140-5p expression as there are no Hif-2α binding sites in the miR-140-5p promoter region.

Point 4: The details of statistical analysis are missing from the entire manuscript. The authors should include the relevant statistical test used in each figure legend and include a section on statistical analysis in the material and methods section.

Response: We thank the reviewer for the instructive suggestion. We are sorry for not describing the statistical analysis in the manuscript. It was missed by mistake and we apologize for the same. A section on statistical analysis in the material and methods section has been added. We have also revised the entire manuscript with the relevant statistical test used in each figure legend.

Point 5: The IHC should be quantified in Fig 8E. Also, the authors should perform IHC for Nrf2.

Response: We thank the reviewer for the valuable comment. Based on your and other reviewer's suggestions, older IHC images are replaced with new clear IHC images taken at 60x magnification. IHC images were quantified. Also, Nrf2 IHC images were added in the revised manuscript.

Point 6: The unprocessed full blots should be provided as supplementary data.

Response: We thank the reviewer for the valuable suggestion. Although we have already submitted all uncropped western blot images at the time of initial manuscript submission, we are resubmitting all western blot images for your consideration.

Point 7: The terms in vitro and in vivo should be italicized in the manuscript.

Response: We thank the reviewer for the instructive suggestion. Terms in vitro and in vivo are italicized and revised in the entire manuscript according to your suggestion.

Reviewer 2 Report

In their manuscript, Mahajan and Sitasawad report the identification of NRF2 as a direct target of miR-140-5p and how miR140-5p inhibits hypoxia-mediated angiogenesis and metastasis through the control of the Nrf2/HO-1 axis.

The paper is well structured and written, and reports a huge amount of results.

However, I have some concerns:

In the Materials and Methods section, the methods of flow cytometry and statistical analysis are missing.

The authors performed their initial experiments on MCF7 and MAD-MB231 cells. However, they do not explain why the second part of the manuscript focuses only on MDA-MB231. Please, clarify this point.

In paragraph 3.2, line 201; the authors stated that knockdown of miR-140-5p, under normoxia, elevated Nrf2 at both mRNA and protein levels but, mRNA is not significantly modified.

In paragraph 3.3, the authors should indicate how the HIF-1α activity was carried out and not only in the caption of Figure 3.

In Figure 4A, rotate the histogram horizontally for better understanding.

In Figures 5D and 5E, what do you mean by the black rectangle? In addition, what does Suramin shown in the hypoxia histogram as a legend for the black box means?

In Figures 5D and 5E, clarify what the division in primary, secondary… etc. means. Do they indicate the number of primary, secondary… etc. blood vessels? Please, specify.

In the paragraph relative to the studies on animals, the authors stated that tumor weight and volume were higher in the miR-140-5p-KD group, but changes in tumor weight were not significantly modified.

Please, provide better images for IHC analysis, the ones shown are blurry, and, in addition, the scale bars are not visible.

In Figure S6, the images for miR-140-5p-OE and tumors are missing.

Minor points:

There are some typos.

Pay attention to the subheading numbering from page 9 to page 17.

Author Response

We would like to express our deepest gratitude to the reviewers for their insightful and valuable comments to improve the scientific value of our manuscript. They really emphasized important issues related with our study. Question wise clarification has been given below.

Reviewer # 2.

In their manuscript, Mahajan and Sitasawad report the identification of NRF2 as a direct target of miR-140-5p and how miR140-5p inhibits hypoxia-mediated angiogenesis and metastasis through the control of the Nrf2/HO-1 axis.

The paper is well structured and written, and reports a huge amount of results.

Thank you very much for your attention to our paper. Based on your suggestions, we have revised the manuscript thoroughly.

Point 1: In the Materials and Methods section, the methods of flow cytometry and statistical analysis are missing.

Response: Since we have given the appropriate references in the materials and methods section for the immunostaining (Hif-1α), apoptosis assay, and cell cycle analysis using flow cytometry, we have not given the detailed protocol for the same.

Point 2: The authors performed their initial experiments on MCF7 and MAD-MB231 cells. However, they do not explain why the second part of the manuscript focuses only on MDA-MB231. Please, clarify this point.

Response: Initially to assess the expression of miRNAs and NRF2, we used both MCF-7 and MDA-MB-231 cells. However, since we had to understand the role of miR-140-5p in breast cancer progression (angiogenesis, migration, and invasion) and MCF-7 cells do not usually migrate or invade (Gest et al., 2013), we carried out experiments using more invasive MDA-MB-231 cells.

Point 3: In paragraph 3.2, line 201; the authors stated that knockdown of miR-140-5p, under normoxia, elevated Nrf2 at both mRNA and protein levels but, mRNA is not significantly modified.

Response: We thank the reviewer for the valuable comment. Knockdown of miR-140-5p under normoxia elevated Nrf2 at both mRNA and protein levels. However, although there is an increase in the mRNA level, it is not seen significantly in the bar graph due to high standard deviation.

Point 4: In paragraph 3.3, the authors should indicate how the HIF-1α activity was carried out and not only in the caption of Figure 3.

Response: We thank you for the valuable suggestion. We apologize for the mistake in writing activity instead of expression. We have made the appropriate change in the revised manuscript.

Point 5: In Figure 4A, rotate the histogram horizontally for better understanding.

Response: As per the reviewer's suggestion, the histogram in Figure 4A has been rotated in the revised manuscript.

Point 6: In Figures 5D and 5E, what do you mean by the black rectangle? In addition, what does Suramin shown in the hypoxia histogram as a legend for the black box means?

Response: The black rectangles in Figure 5D and 5E represent VEGF and Suramin, respectively. VEGF and Suramin were used as positive and negative controls respectively, in the Yolk Sac Membrane (YSM) assay. We have also mentioned the role of VEGF and Suramin in the legend of Figure 5.

Point 7: In Figures 5D and 5E, clarify what the division in primary, secondary… etc. means. Do they indicate the number of primary, secondary… etc. blood vessels? Please, specify.

Response: Since angiogenesis is the formation of new blood vessels from pre-existing ones by sprouting, branching, and differential growth of blood vessels (Medhora, M. et al., 2008), the terms primary, secondary, tertiary, and quaternary refer to the branching of blood vessels.

Point 8: In the paragraph relative to the studies on animals, the authors stated that tumor weight and volume were higher in the miR-140-5p-KD group, but changes in tumor weight were not significantly modified.

Response: We thank the reviewer for the valuable comment. Tumor size and weight are more in the miR-140-5p-KD (knockdown) group. The differences were not statistically significant. miR-140-5p is functioning as a tumor suppressor. Overexpression of miR-140-5p potently suppressed tumor growth, while knockdown of miR-140-5p recovered tumor growth. If we compare empty vector (EV) injected mice with the highest tumor weight with tumors from miR-140-5p-KD mice with the least tumor weight, it increases.

Point 9: Please, provide better images for IHC analysis, the ones shown are blurry, and, in addition, the scale bars are not visible.

Response: As per the reviewer's suggestion, we have replaced the older IHC images with the new clear IHC images taken at 60x magnification in the revised manuscript.

Point 10: In Figure S6, the images for miR-140-5p-OE and tumors are missing.

Response: In our study, we have reported miR-140-5p as a tumor suppressor and since the tumor growth is potently suppressed in the miR-140-5p-OE group compared to EV, we did not find any tumor in the miR-140-5p-OE group. Therefore, tumor images from the miR-140-5p-OE group were not shown in Figure S6.

Minor points:

There are some typos.

Pay attention to the subheading numbering from page 9 to page 17.

As per the reviewer's suggestion, the numbering throughout the manuscript has been corrected and incorporated into the revised manuscript.

References:

Gest, C., Joimel, U., Huang, L., Pritchard, L. L., Petit, A., Dulong, C., ... & Soria, C. (2013). Rac3 induces a molecular pathway triggering breast cancer cell aggressiveness: differences in MDA-MB-231 and MCF-7 breast cancer cell lines. BMC cancer, 13(1), 1-14.

Medhora, M., Dhanasekaran, A., Pratt Jr, P. F., Cook, C. R., Dunn, L. K., Gruenloh, S. K., & Jacobs, E. R. (2008). Role of JNK in network formation of human lung microvascular endothelial cells. American Journal of Physiology-Lung Cellular and Molecular Physiology, 294(4), L676-L685.

Reviewer 3 Report

The study investigated the involvement of epigenetic mechanism ( micro-RNAs) in hypoxia-related breast cancer (BC) progression. The authors found that  miR-140-5p is downregulated in BC cells under hypoxic conditions. They also found that Nrf2 is a direct target of miR-140-5p. The study is interesting and properly designed. Overall, the paper is well-written, although some English spell-/grammar check is required. There are also several issues to address.

  1. Abstract: all abbreviations should be properly deciphered. Line11: NF-E2; should be “ nuclear factor erythroid 2‑related factor 2 (Nrf2)”.
  2. Lines 17-19 : it is better to break this long sentence into 2 ( Knockdown of miR-140-5p under normoxic conditions significantly enhanced while its overexpression under hypoxic conditions abrogated Nrf2/HO-1 signaling and tumor growth, angiogenesis, migration, and invasion in BC).
  3. In vivo study data should be indicated in the abstract.
  4. Introduction: few more references from 2020-2021 (check MDPI Special Issues) can be included. It is necessary to indicate recent findings about expression of Nrf2 in breast cancer cells ( see this https://pubmed.ncbi.nlm.nih.gov/34572304/)
  5. Line 42: Authors wrote “Nrf2… a novel chemotherapeutic strategy for treating solid tumors”. However, Nrf2 is not a very novel target. Replace “novel” with “promising” etc.
  6. Discussion: compare your findings for Nrf2 expression levels with recently published data (Bovilla et al., 2021; ).
  7. Line 206: replace “ from studying” with “ to study…”
  8. Line 238: should be “ regulated independently…”
  9. Figure 6G; EV-is empty vector; what is “Control” stands for in the last blot? Was is EV or vehicle-treated cells? Clarify.
  10. Figure 7D: what is “SC” and how cells were treated? Indicate.
  11. Figure 8E. IHC staining in the xenograft tumors -20x Objective lens is very low magnifications. It is impossible to see cells and tissue structure. Higher magnification (60X or 100X) images should be included.
  12. Discussion: the link between miR-140/oxidative stress and Nrf2 was not covered in full details; some key-references were not included /discussed (https://pubmed.ncbi.nlm.nih.gov/30483753/ ; https://pubmed.ncbi.nlm.nih.gov/30483753/ ).

Author Response

We would like to express our deepest gratitude to the reviewers for their insightful and valuable comments to improve the scientific value of our manuscript. They really emphasized important issues related with our study. Question wise clarification has been given below.

Reviewer # 3.

The study investigated the involvement of epigenetic mechanism ( micro-RNAs) in hypoxia-related breast cancer (BC) progression. The authors found that  miR-140-5p is downregulated in BC cells under hypoxic conditions. They also found that Nrf2 is a direct target of miR-140-5p. The study is interesting and properly designed. Overall, the paper is well-written, although some English spell-/grammar check is required.

Thank you very much for your attention to our paper. Based on your suggestions, we have revised the manuscript thoroughly.

Point 1: Abstract: all abbreviations should be properly deciphered. Line11: NF-E2; should be “nuclear factor erythroid 2‑related factor 2 (Nrf2)”.

Response: We thank the reviewer for the valuable suggestion. All the abbreviations are appropriately written in the revised manuscript.

Point 2: Lines 17-19 : it is better to break this long sentence into 2 ( Knockdown of miR-140-5p under normoxic conditions significantly enhanced while its overexpression under hypoxic conditions abrogated Nrf2/HO-1 signaling and tumor growth, angiogenesis, migration, and invasion in BC).

Response: We thank the reviewer for the valuable suggestion. The sentence has been modified in lines 17-19 according to the suggestion in the revised manuscript.

Point 3: In vivo study data should be indicated in the abstract.

Response: We thank the reviewer for the valuable comment. The in vivo data has been indicated in the revised manuscript.

Point 4: Introduction: few more references from 2020-2021 (check MDPI Special Issues) can be included. It is necessary to indicate recent findings about expression of Nrf2 in breast cancer cells (see this https://pubmed.ncbi.nlm.nih.gov/34572304/)

Response: We thank the reviewer for the important suggestion. We have added the related references in the revised manuscript. In line with these changes, we have also described Nrf2 expression in breast cancer cells.

Point 5: Line 42: Authors wrote “Nrf2… a novel chemotherapeutic strategy for treating solid tumors”. However, Nrf2 is not a very novel target. Replace “novel” with “promising” etc.

Response: We thank the reviewer for the valuable suggestion. We modified line 42 accordingly in the revised manuscript.

Point 6: Discussion: compare your findings for Nrf2 expression levels with recently published data (Bovilla et al., 2021;).

Response: As per the reviwer's suggestion, changes have been made in the revised manuscript.

Point 7: Line 206: replace “from studying” with “to study…”

Response: Modification in the line 206 have been made as per the reviewer's suggestion,

Point 8: Line 238: should be “regulated independently…”

Response:  We accept the reviwer's suggestion, and modification has been made on line 238.

Point 9: Figure 6G; EV-is empty vector; what is “Control” stands for in the last blot? Was is EV or vehicle-treated cells? Clarify.

Response: Special thanks to the reviewer for this comment. In Figure 6G and also 7G, it was a mistake. “Control” should be SC (Scrambled) and modifications have been made in both the figures. In SC group, cells were stably transfected with scrambled shRNA vector and used as a negative control.

Point 10: Figure 7D: what is “SC” and how cells were treated? Indicate.

Response: SC stands for scrambled shRNA. Cells were stably transfected with lentiviruses expressing scrambled shRNA vector and are used as negative control.

Point 11: Figure 8E. IHC staining in the xenograft tumors -20x Objective lens is very low magnifications. It is impossible to see cells and tissue structure. Higher magnification (60X or 100X) images should be included.

Response: We thank the reviewer for the valuable comment. Based on the suggestion, older IHC images were replaced with new clear IHC images taken at 60x magnification in the revised manuscript.

Point 12: Discussion: the link between miR-140/oxidative stress and Nrf2 was not covered in full details; some key-references were not included /discussed (https://pubmed.ncbi.nlm.nih.gov/30483753/; https://pubmed.ncbi.nlm.nih.gov/30483753/).

Response: We thank the reviewer for the valuable suggestion. Based on your suggestion, the reference has been incorporated in the revised manuscript.  

Round 2

Reviewer 1 Report

The authors have addressed the majority of the concerns satisfactorily. Although two comments have not been addressed fully citing time constraints, I believe, they might not have a significant impact on the findings of the study and would have strengthened the findings further. The manuscript in the current form still holds great value and should be accepted.

Author Response

Reviewer # 1

The authors have addressed the majority of the concerns satisfactorily. Although two comments have not been addressed fully citing time constraints, I believe, they might not have a significant impact on the findings of the study and would have strengthened the findings further. The manuscript in the current form still holds great value and should be accepted.

We would like to express our deepest gratitude to the reviewers for their insightful and valuable comments to improve the scientific value of our manuscript. They really emphasized important issues related with our study.

Thank you

Reviewer 2 Report

I am not completely satisfied with the revision proposed by the authors.

Point 2

This concept should be reported also in the text of the manuscript.

Point 3 and Point 8

If the variations in expression (mRNA) and weight (in animal experiments) are not significant, means that they are not variations but only oscillation, not statistically significant!

Point 9

The IHC image (figure 8E) is still very blurry and of little use in this way.

Point 10

Even if the animals after treatment showed "complete" tumor inhibition, they should still be shown (Figure S6 A) it is not scientifically correct to show a white rectangle unless the treatment makes the animals disappear too... For panel B of figure S6, the authors should indicate in the caption that the treated animals did not develop tumors, even if potently suppressed tumor growth (as stated by the authors) does not mean completely suppressed tumor growth.

Author Response

Reviewer # 2.

I am not completely satisfied with the revision proposed by the authors.

Thank you very much for your attention to our paper. Based on your suggestions, we have revised the manuscript thoroughly.

Point 2: This concept should be reported also in the text of the manuscript.

Response: We thank the reviewer for the important suggestion. We have made changes accordingly in Paragraph 3.4 in the revised manuscript. We have also added the related references in the revised manuscript.

Point 3 and Point 8: If the variations in expression (mRNA) and weight (in animal experiments) are not significant, means that they are not variations but only oscillation, not statistically significant!

Response: We thank the reviewer for the valuable comment. For Nrf2 mRNA levels in miR-140-5p knockdown cells, we have repeated this experiment again, and the data has been revised accordingly. miR-140-5p knockdown significantly increased Nrf2 mRNA levels.

For tumor weight, we plotted the best three values (tumor weight) as earlier data was from 5 mice, and the data has been revised accordingly.

Point 9: The IHC image (figure 8E) is still very blurry and of little use in this way.

Response: We thank the reviewer for the important suggestion. We replaced earlier IHC images with new images taken at 60x magnification with good clarity.

Point 10: Even if the animals after treatment showed "complete" tumor inhibition, they should still be shown (Figure S6 A) it is not scientifically correct to show a white rectangle unless the treatment makes the animals disappear too... For panel B of figure S6, the authors should indicate in the caption that the treated animals did not develop tumors, even if potently suppressed tumor growth (as stated by the authors) does not mean completely suppressed tumor growth.

Response: We thank the reviewer for the important suggestion. Based on your suggestion, mice from miR-140-5p overexpression group were shown in Figure S6A. Also, Figure S6 caption has been revised as per your suggestion.

Reviewer 3 Report

I am satisfied with the revised version of he manuscript. Authors addressed all my comments.

Author Response

Reviewer # 3

I am satisfied with the revised version of the manuscript. Authors addressed all my comments.

We would like to express our deepest gratitude to the reviewers for their insightful and valuable comments to improve the scientific value of our manuscript. They really emphasized important issues related with our study.

Thank you
